DATA RELEASE

# The Crown Pearl V2: an improved genome assembly of the European freshwater pearl mussel *Margaritifera margaritifera* (Linnaeus, 1758)

André Gomes-dos-Santos[1,2,*], Manuel Lopes-Lima[3,4], André M. Machado[1], Thomas Forest[5,6,7], Guillaume Achaz[5,6], Amílcar Teixeira[8], Vincent Prié[3,4], L. Filipe C. Castro[1,2] and Elsa Froufe[1,*]

1　CIIMAR/CIMAR - Interdisciplinary Centre of Marine and Environmental Research, University of Porto, Matosinhos, Portugal
2　Faculty of Sciences, University of Porto, , Porto, Portugal
3　CIBIO/InBIO - Research Center in Biodiversity and Genetic Resources, University of Porto, Vairão, Portugal
4　IUCN SSC Mollusc Specialist Group, c/o IUCN, David Attenborough Building, Pembroke St., Cambridge, England
5　Éco-anthropologie, Muséum National d'Histoire Naturelle, CNRS UMR 7206, Paris, France
6　SMILE Group, Center for Interdisciplinary Research in Biology (CIRB), Collège de France, CNRS UMR 7241, INSERM U 1050, Paris, France
7　Institut de Systématique Evolution Biodiversité, CNRS MNHN SU EPHE, CP 51, 55 rue Buffon, 75005, Paris, France
8　Centro de Investigação de Montanha (CIMO), Instituto Politécnico de Bragança, Bragança, Portugal

**Submitted:**　09 February 2023

* Corresponding authors. E-mail: andrepousa64@gmail.com; elsafroufe@gmail.com

Preprint submitted at https://doi.org/10.1101/2023.02.11.528107

## ABSTRACT

Contiguous assemblies are fundamental to deciphering the composition of extant genomes. In molluscs, this is considerably challenging owing to the large size of their genomes, heterozygosity, and widespread repetitive content. Consequently, long-read sequencing technologies are fundamental for high contiguity and quality. The first genome assembly of *Margaritifera margaritifera* (Linnaeus, 1758) (Mollusca: Bivalvia: Unionida), a culturally relevant, widespread, and highly threatened species of freshwater mussels, was recently generated. However, the resulting genome is highly fragmented since the assembly relied on short-read approaches. Here, an improved reference genome assembly was generated using a combination of PacBio CLR long reads and Illumina paired-end short reads. This genome assembly is 2.4 Gb long, organized into 1,700 scaffolds with a contig N50 length of 3.4 Mbp. The *ab initio* gene prediction resulted in 48,314 protein-coding genes. Our new assembly is a substantial improvement and an essential resource for studying this species' unique biological and evolutionary features, helping promote its conservation.

**Subjects**　Genetics and Genomics, Animal Genetics, Freshwater Science

## DATA DESCRIPTION

### Background and context

Initial efforts to sequence molluscan genomes relied primarily on short-read approaches, which, despite their unarguable value, frequently result in highly fragmented

assemblies [1–4]. Consequently, long-read sequencing approaches, such as Pacific Bioscience (PacBio) or Oxford Nanopore Technology, are becoming the common ground of emerging studies of molluscan genome assemblies [1–4]. This is further facilitated by the decreasing cost trend, coupled with the increasing sequencing accuracy of these approaches [5]. Additionally, the structural information provided by long-reads is crucial to span large indels or inform about long structural variants [6–8], which is particularly relevant for molluscans that have large, heterozygous, and highly repetitive genomes (reviewed in [4]). Consequently, long-read-based reference assemblies have reduced fragmentation levels, fewer missing and truncated genes, and reduced chances of chimerically assembled regions [6, 7].

Bivalves from the order Unionida, commonly known as freshwater mussels, are the most diverse group of strictly freshwater bivalves, with over 1,000 species distributed across all continents except Antarctica [9, 10]. The freshwater pearl mussel *Margaritifera margaritifera* (Linnaeus, 1758) (NCBI:txid2505931) is perhaps the most emblematic, culturally significant, and known species of freshwater mussels. The freshwater pearl mussel is also the only species of the group that inhabits both European and North American freshwater systems [11, 12] (Figure 1), mainly cool oligotrophic waters. Moreover, this species holds a series of distinctive biological features, such as the ability to produce pearls (with an ancient history of pearl harvesting [13, 14]), a long lifespan (reaching over 200 years [15]) with negligible signs of cellular senescence [16], and, as all other freshwater mussels, an obligatory parasitic life stage on salmonid fish species [12, 17]. In the past, the freshwater pearl mussel was highly abundant across its Holarctic distribution [12]. However, during the last century, the species has suffered massive declines due to the many human-mediated threats impacting the freshwater ecosystems [11, 12]. As a result, the species is listed as critically endangered in Europe and included in the European Habitats Directive under Annexes II and V, and the Appendix III of the Bern Convention [11].

Despite the cultural significance and poor conservation status of the freshwater pearl mussel, the availability of genomic resources to study this species is still limited [13, 18–22]. Also, almost nothing is known about the molecular mechanism governing the regulation and functioning of its many relevant biological features. Genomic resources provide benchmarking tools to monitor, identify, and classify conservation units as well as classify genetic elements with conservation relevance and adaptive potential [23, 24]. Hence, genomics provides invaluable tools to improve the success of conservation efforts. The sequencing of the first genome for the freshwater pearl mussel represented a fundamental resource for the study of its biology and evolution and, ultimately, promoted its conservation [13]. However, although the quality of this first assembly is good (validated with several statistics), it was produced using solely short-read sequencing (i.e., Illumina paired-end and mate-pair sequencing), thus hampering its overall contiguity [13]. The subsequent release of the highly contiguous genome assembly of the freshwater mussel *Potamilus streckersoni* [25], which relied on PacBio long-read sequencing, demonstrated how using longer reads is critical to ensure improved contiguity of genome assemblies to study freshwater mussels [26].

In this study, we aimed to improve the genome assembly of the freshwater pearl mussel *M. margaritifera*. Therefore, the genome of a new individual from this species was sequenced using PacBio CLR and Illumina paired-end short reads. As a result, we generated the most contiguous genome assembly of freshwater mussels available to date, significantly improving its contiguity and completeness [13].



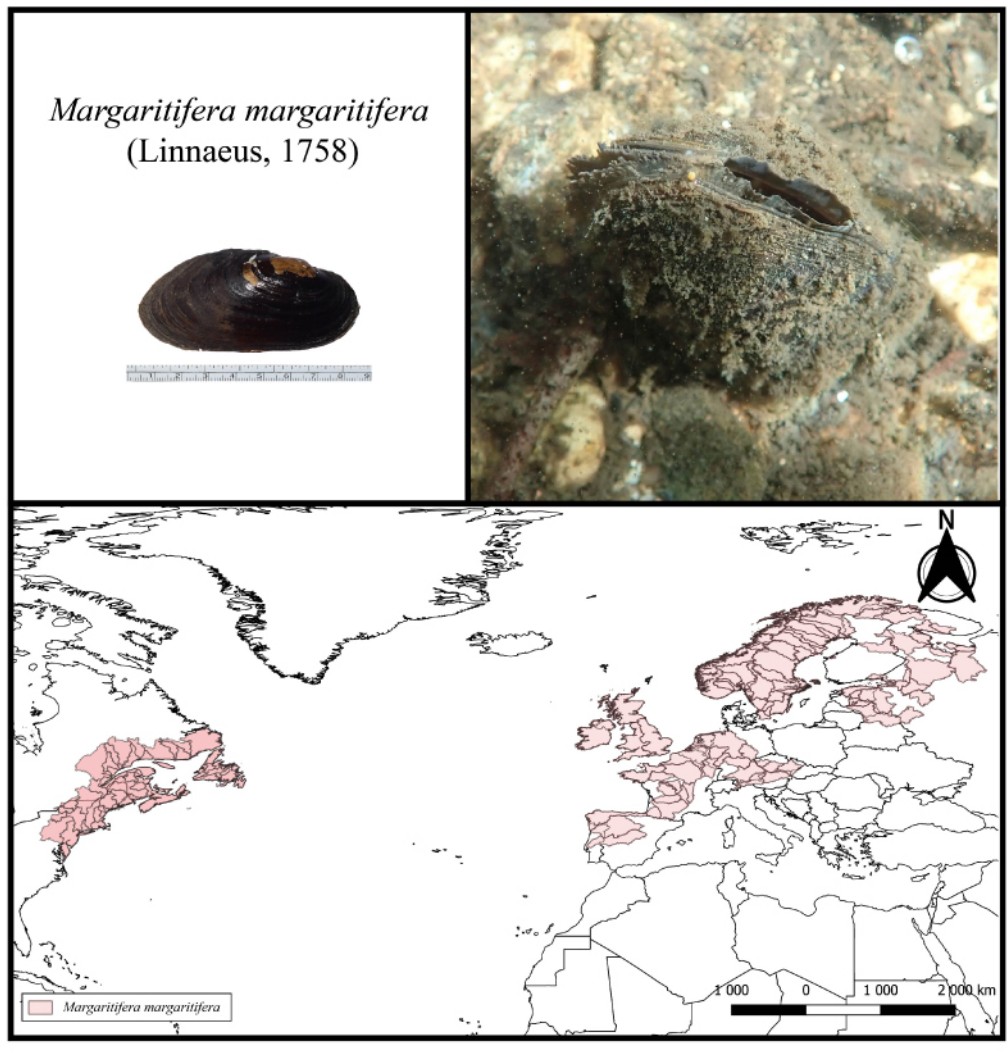

**Figure 1.** Top left: The *M. margaritifera* specimen used for the whole genome assembly of this study. Top right: A specimen of *M. margaritifera* in its natural habitat (Photos by André Gomes-dos-Santos). Bottom: Map of the potential distribution of the freshwater pearl mussel, produced by overlapping points of recent presence records [11] with Hydrobasins level 5 polygons [27]. The potential distribution for Europe was retrieved from [11] and for North America from [28].

## METHODS

### Animal sampling

One individual of *M. margaritifera* was collected from the Tuela River in Portugal (Table 1) and transported alive to the laboratory, where tissues were separated, flash-frozen, and stored at −80 °C. The shell and tissues are deposited in the CIIMAR tissue and mussels' collection.

### DNA extraction and sequencing

For the PacBio sequencing, the mantle tissue was sent to Brigham Young University (BYU, USA). High-molecular-weight DNA extraction was performed, and PacBio library construction was achieved following the single-molecule real-time (SMRT) bell construction

**Table 1.** Sample details for the freshwater pearl mussel *M. margaritifera* specimen used for our whole genome sequencing (WGS).

| Sample | *Margaritifera margaritifera* |
|---|---|
| Investigation_type | Eukaryote |
| Lat_lon | 41.862414; −6.931596 |
| Geo_loc_name | Portugal |
| Collection_date | 06/07/2021 |
| Env_package | Water |
| Collector | Amílcar Teixeira |
| Sex | Undetermined |
| Maturity | Mature |

**Table 2.** General statistics of the raw sequencing reads used for the *M. margaritifera* genome assembly.

| Sample | Sequencing type | Library type | Platform | Insert size (bp) | Number of reads | Application |
|---|---|---|---|---|---|---|
| PacBio CLR | WGS | Long reads | PacBio Sequel II system | 14,339 | 7,892,056 | Genome assembly, Genome polishing |
| Illumina PE | WGS | Short reads | Novaseq 6000 | 450 | 1,369,564,530 | Genome size estimation, Genome polishing |

protocol [29]. The library was sequenced using an SMRT cell of a PacBio Sequel II system v.9.0. The genomic DNA for short-read sequencing was extracted from the muscle tissue using the Qiagen MagAttract HMW DNA Kit, following the manufacturer's instructions. The extracted DNA was sent to Macrogen Inc. for standard Illumina Truseq Nano DNA library preparation, and the WGS of 150 bp paired-end reads on the Illumina Novaseq 6,000 machine (Table 2).

## Genome assembly and annotation
The overall pipeline used to obtain the genome assembly and annotation is provided in Figure 2.

## Genome size and heterozygosity estimation
Before the assembly, the characteristics of the genome were accessed with a k-mer frequency spectrum using the paired-end reads. First, the quality of the reads was evaluated using FastQC (v.0.11.8; RRID:SCR_014583) [30]. The reads were then quality trimmed with Trimmomatic (v.0.38; RRID:SCR_011848) [31], specifying the parameters "LEADING: 5 TRAILING: 5 SLIDINGWINDOW: 5:20 MINLEN: 36". Finally, the quality of the clean reads was validated using FastQC and then used for the genome size estimation with Jellyfish (v2.2.10; RRID:SCR_005491) and GenomeScope2 [32], specifying the k-mer length of 21.

## Genome assembly
The primary genome assembly was constructed using the raw PacBio reads with NextDenovo v2.4.0 [33], with default parameters and specifying an estimated genome size of 2.4 Gbp. Polishing of the resulting assembly was performed in two steps. First, we used the PacBio reads with three iterations of GCpp v2.0.2 [34], and then we used the clean paired-end reads with two iterations of NextPolish v1.2.3 [35]. Specifically, the PacBio read alignments were performed with pbmm2 v1.4.0 [36], and the paired-end read alignments were performed with Burrows-Wheeler Aligner (BWA; v0.7.17; RRID:SCR_010910) [37], both with default parameters.



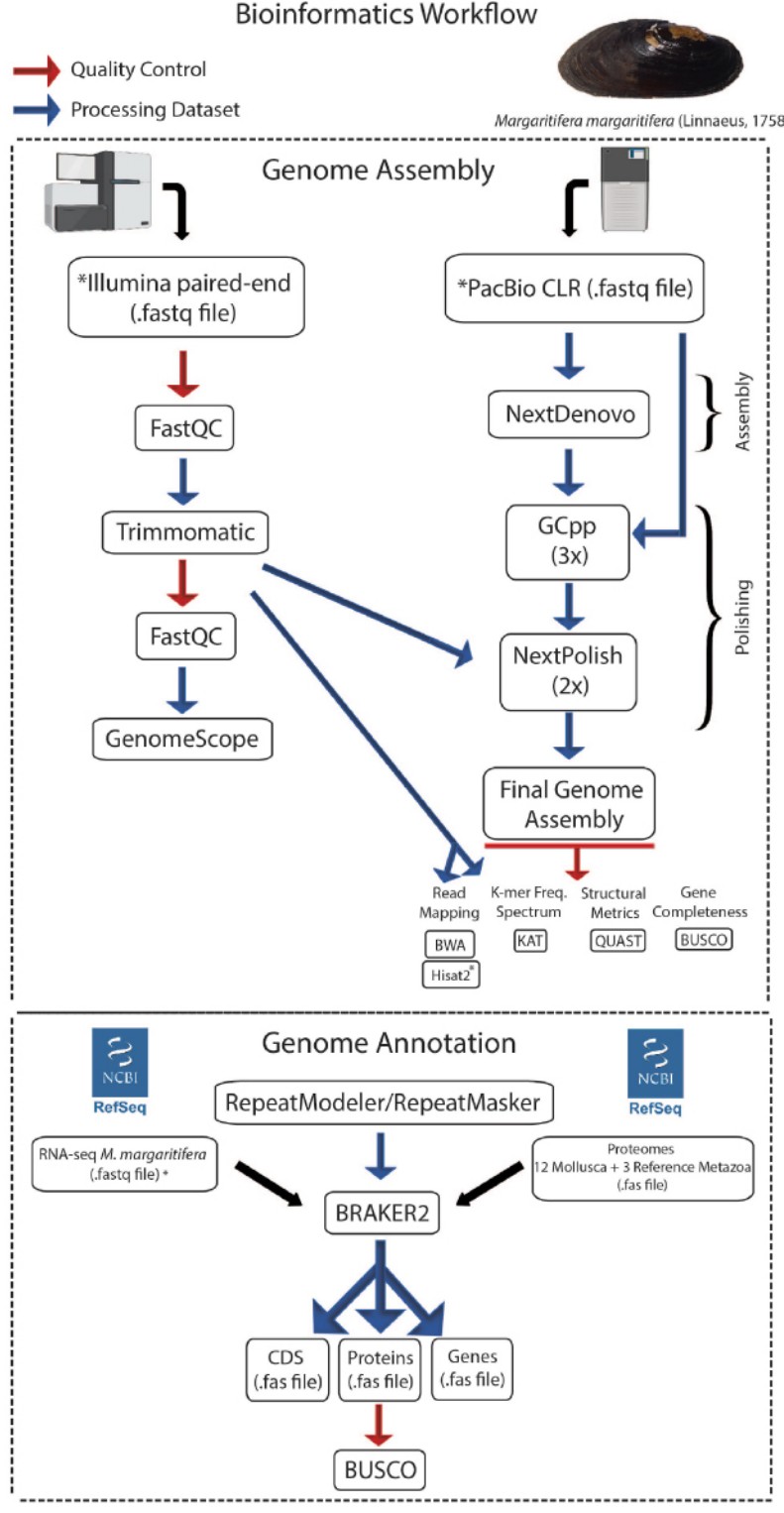

**Figure 2.** Bioinformatics pipeline used for the genome assembly and annotation.

**Table 3.** List of the proteomes used for the BRAKER2 gene prediction pipeline.

| Phylum | Class | Order | Species | GenBank/RefSeq |
|---|---|---|---|---|
| Mollusca | Bivalvia | | | |
| | | Ostreida | *Crassostrea gigas* | GCF_902806645.1 |
| | | | *Crassostrea virginica* | GCF_002022765.2 |
| | | Pectinida | *Mizuhopecten yessoensis* | GCF_000457365.1 |
| | | | *Pecten maximus* | GCF_902652985.1 |
| | | Veneroida | *Dreissena polymorpha* | GCF_020536995.1 |
| | | | *Mercenaria mercenaria* | GCF_014805675.1 |
| | | Unionida | *Margaritifera margaritifera* | GCA_015947965.1 |
| | | | *Megalonaias nervosa* | GCA_016617855.1 |
| | Gastropod | | *Biomphalaria glabrata* | GCF_000457365.1 |
| | | | *Pomacea canaliculata* | GCF_003073045.1 |
| | | | *Gigantopelta aegis* | GCF_016097555.1 |
| | Cephalopod | | *Octopus bimaculoides* | GCF_001194135.1 |
| | | | *Octopus sinensis* | GCF_006345805.1 |
| | Polyplacophora | | *Acanthopleura granulata* | GCA_016165875.1 |
| Chordata | | | *Homo sapiens* | GCF_000001405.40 |
| Chordata | | | *Ciona intestinalis* | GCF_000224145.3 |
| Echinodermata | | | *Strongylocentrotus purpuratus* | GCF_000002235.4 |

The general statistics and completeness of the final genome assembly were estimated with QUAST (v5.0.2; RRID:SCR_001228) [38], BUSCO (v5.2.2; RRID:SCR_015008) [39], and using the paired-end reads for read-back mapping with BWA, and a k-mer frequency distribution analysis with the K-mer Analysis Toolkit (KAT) [40].

## Masking of repetitive elements, gene models' predictions, and annotation

To mask repetitive elements, a *de novo* library of repeats was created for final genome assembly with RepeatModeler (v2.0.133; RRID:SCR_015027) [41]. Next, the genome was soft masked with RepeatMasker (v4.0.734; RRID:SCR_012954) [42], combining the *de novo* library with the 'Bivalvia' libraries from Dfam [43] (Dfam_consensus-20170127) and RepBase [44] (RepBaseRepeatMaskerEdition-20181026).

Gene prediction was performed on the soft-masked genome assembly using the BRAKER2 pipeline v2.1.6 [45]. First, all the available RNA-seq data of *M. margaritifera* from GenBank [22, 46] and Gomes-dos-Santos *et al.* [18] (the latter used the same *M. margaritifera* individual used for the genome assembly of this study) was retrieved and quality trimmed with Trimmomatic v.0.38 (parameters described above). Next, the clean reads were aligned to the masked genome using Hisat2 (v.2.2.0; RRID:SCR_015530) with the default parameters [47]. Furthermore, the complete proteomes of 14 mollusc species and three reference Metazoan species (*Homo sapiens*, *Ciona intestinalis*, *Strongylocentrotus purpuratus*), downloaded from public databases (Table 3), were used as additional evidence for gene prediction. The BRAKER2 pipeline was then applied, specifying the parameters "–etpmode; –softmasking;". Gene predictions were renamed (Mma), cleaned, and filtered using AGAT v.0.8.0 [48], correcting overlapping predictions and removing incomplete gene predictions (i.e., without start and/or stop codons). Finally, proteins were extracted from the genome using AGAT, and a functional annotation was performed using InterProScan (v.5.44.80; RRID:SCR_005829) [49] and BLASTP (RRID:SCR_001010) searches against the RefSeq database [50]. Homology searches were performed using DIAMOND (v.2.0.11.149;



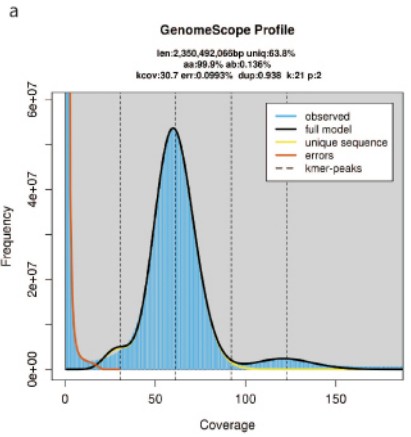
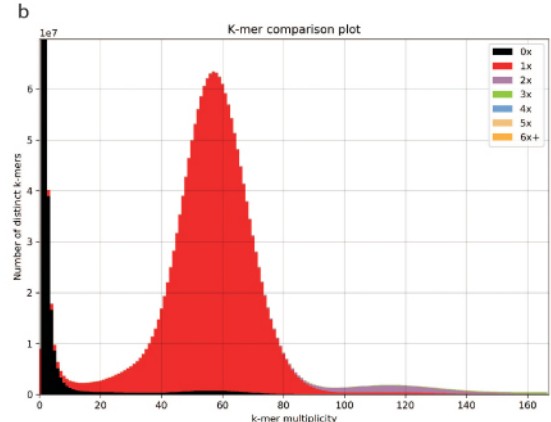

**Figure 3.** (a) GenomeScope2 k-mer () distribution displaying the estimation of the genome size (len), the homozygosity (aa), the heterozygosity (ab), the mean coverage of k-mer for heterozygous bases (kcov), the read error rate (err), the average rate of read duplications (dup), the size of the k-mer used on the run (k), the ploidy (p), and percentage of the genome that is unique (not repetitive) (uniq). (b) *M. margaritifera* genome assembly assessment using the KAT comp tool to compare the Illumina paired-end k-mer content within the genome assembly. Different colours represent the read k-mer frequency in the assembly.

RRID:SCR_016071) [51], specifying the parameters "–k 1, –b 20, –e 1e-5, –sensitive, –outfmt 6". Finally, BUSCO scores were estimated for the predicted proteins [39].

## DATA VALIDATION

### Sequencing results and genome assembly

The raw sequencing outputs resulted in 103 Gbp of raw PacBio and 203 Gbp of raw paired-end reads. A total of 201 Gbp of paired-end reads were maintained after trimming and quality filtering. Similarly to the results of Gomes-dos-Santos et al. [13], the GenomeScope2 estimated genome size was ~2.36 Gb, and the heterozygosity levels were low, i.e., ~0.163% (Figure 3a).

The final genome assembly (hereafter referred to as Genome V2) has a total size of 2.45 Gbp, similar to the genome size reported in a previous assembly [13] (hereafter referred to as Genome V1). Regarding the contiguity, Genome V2 shows a contig N50 of 3.42 Mbp (Table 4), representing a ~202-fold increase in contig N50 and ~11-fold increase in scaffold N50 relative to Genome V1 (Table 4). Additionally, Genome V2 represents the most contiguous freshwater mussel genome assembly currently available [13, 26, 52, 53]. Genome V2 shows a ~1.66-fold increase in N50 length compared to the other PacBio-based genome assembly, i.e., from *P. streckersoni* [26]. This observation is striking considering that the Genome V2 is larger (nearly 4 Mbp longer), has more repetitive elements (nearly 7% more) (Tables 4 and 5) and similar heterozygosity (nearly 0.43% less) (Figure 3).

Genome V2 also shows a considerable increase in the BUSCO scores, with nearly no fragmented nor missing hits for both the eukaryotic and metazoan curated lists of near-universal single-copy orthologous genes (Table 4). Short-read back-mapping percentages resulted in an almost complete read mapping and a 99.69% alignment rate (Table 4). The KAT k-mer distribution spectrum revealed that almost all read information was included in the final assembly (Figure 3b). Overall, these general statistics validate the high completeness, low redundancy, and quality of the Genome V2.

**Table 4.** General statistics of the two *M. margaritifera* genome assemblies, including read alignment, gene prediction, and annotation.

| | Genome V2 contig* | Genome V1 contig | Genome V1 scaffold | *Megalonaias nervosa* | *Potamilus streckersoni* | *Venustaconcha ellipsiformis* |
|---|---|---|---|---|---|---|
| Total number of sequences ≥ 1,000 bp | 1,700 | 265,718 | 105,185 | 90,895 | 2,366 | 371,427 |
| Total number of sequences ≥ 10,000 bp | 1,700 | 66,019 | 15,384 | 54,764 | 2,162 | 26,952 |
| Total number of sequences ≥ 25,000 bp | 1,202 | 18,725 | 11,583 | 29,042 | 1,831 | 5,073 |
| Total number of sequences ≥ 50,000 bp | 1,570 | 4,284 | 9,265 | 12,699 | 1,641 | 1,456 |
| Total length ≥ 1,000 bp | 2.45 Gb | 2.2 Gb | 2.47 Gb | 2.36 Gb | 1.77 Gb | 1.59 Gb |
| Total length ≥ 10,000 bp | 2.45 Gb | 1.52 Gb | 2.29 Gb | 2.19 Gb | 1.77 Gb | 0.54 Gb |
| Total length ≥ 25,000 bp | 2.45 Gb | 789 Mb | 2.23 Gb | 1.76 Gb | 1.76 Gb | 0.23 Gb |
| Total length ≥ 50,000 bp | 2.44 Gb | 299 Mb | 2.15 Gb | 1.19 Gb | 1.76 Gb | 0.10 Gb |
| N50 length (bp) | 3.42 Mb | 16 Kb | 288 Kb | 50 Kb | 2.05 Mb | 6,657 |
| L50 | 207 | 34,910 | 2,393 | 12,463 | 245 | 58,531 |
| Largest contig (bp) | 23 Mb | 0.209 Mb | 2.5 Mb | 0.588 Mb | 10 Mb | 313,274 |
| GC content, % | 35.3 | 35.42 | 35.42 | 35.82 | 33.79 | 34.19 |
| Clean paired-end (PE) Reads Alignment Stats | | | | | | |
| Percentage of Mapped RNA-seq PE (%) | - | Average 96.94 | - | 97.75 | - | - |
| Percentage of Mapped WGS PE (%) | - | 99.69 | - | 97.75 | - | - |
| Total BUSCO for the genome assembly (%) | | | | | | |
| #Euk database | - | C:99.2% [S:97.6%, D:1.6%], F:0.4% | - | C:86.8% [S:85.8%, D:1.0%], F:5.9% | C:70.6% [S:70.2%, D:0.4%], F:14.9% | C:98.1% [S:97.3%, D:0.8%], F:0.8% | C:45.9% [S:45.5%, D:0.4%], F:36.9% |
| #Met database | - | C:96.9% [S:95.5%, D:1.4%], F:2.0% | - | C:84.9% [S:83.8%, D:1.1%], F:4.9% | C:71.5% [S:70.1%, D:1.4%], F:14.5% | C:95.0% [S:93.6%, D:1.4%], F:2.3% | C:53.7% [S:52.8%, D:0.9%], F:29.7% |
| Masking Repetitive Regions and Gene Prediction | | | | | | |
| Percentage masked bases (%) | - | 57.32 | - | 59.07 | 25.00 | 51.03 | 36.29 |
| Number of mRNAs | - | 48,314 | - | 40,544 | 49,149 | 41,065 | 41,697 |
| Protein coding genes (CDS) | - | 48,314 | - | 35,119 | 49,149 | 41,065 | - |
| Functional annotated genes | | 35,649 | - | 31,584 | - | - | - |
| Total gene length (bp) | - | 1.13 Gb | - | 902 Mb | - | - | - |
| Total BUSCO for the predicted proteins (%) | | | | | | |
| + Euk database | - | C:97.6% [S:83.9%, D:13.7%], F:2.0% | - | C:90.6% (S:81.2%, D:9.4%), F:3.9% | - | - | - |
| + Met database | - | C:98.7% [S:84.7%, D:14.0%], F:0.8% | - | C:92.6% (S:82.3%, D:10.3%), F:3.2% | - | - | - |

*Genome V2 refers to the new assembly here produced and is solely at the contig level, i.e., has no scaffolds; Genome V1 refers to the first *M. margaritifera* genome [13]; #Euk: From a total of 303 genes of Eukaryota library profile; #Met: From a total of 978 genes of Metazoa library profile; + Euk: From a total of 255 genes of Eukaryota library profile; + Met: From a total of 954 genes of Metazoa library profile; #,+ C: Complete; S: Single; D: Duplicated; F: Fragmented.

### Repeat masking, gene models prediction, and annotation

RepeatModeler/RepeatMasker masked 57.32% of Genome V2, 1.75% less than the values reported for Genome V1. This result was likely a consequence of the new assembly being able to resolve repetitive regions more accurately (Table 5). Furthermore, this value was considerably higher than the estimated duplications of GenomeScope, i.e., 36.2% (Figure 3a, Table 5). These differences have been observed in other assemblies of freshwater mussel genomes [4, 26, 52] and are likely due to the inaccurate estimation of repeat content when applying k-mer frequency spectrum analysis in highly repetitive genomes using short reads. Similarly to Genome V1, most repeats in Genome V2 were unclassified (27.26%, ~668 Mgp), followed by DNA elements (17.18%, ~421 Mgp), long terminal repeats (5.95%, ~145 Mgp), long interspersed nuclear elements (5.86%, ~143 Mgp), and short interspersed nuclear elements (0.75%, ~18 Mgp) (Table 5). BRAKER2 gene prediction identified 48,314 CDS, an increase compared with Genome V1 and closer to the predictions of the other two freshwater mussel assemblies (Tables 4 and 6). This result probably reflects the higher contiguity and completeness of Genome V2, as evidenced by the high BUSCO scores for protein predictions, with almost no missing hits for either of the near-universal single-copy

**Table 5.** RepeatMasker report of the content of repetitive elements in the new *M. margaritifera* genome assembly.

|  |  | Number of elements | Length occupies | Percentage of sequence |
|---|---|---|---|---|
| SINEs: |  | 99,204 | 18,473,640 bp | 0.75% |
|  | ALUs | 0 | 0 bp | 0.00% |
|  | MIRs | 48,204 | 8,371,356 bp | 0.34% |
| LINEs: |  | 345,367 | 143,734,167 bp | 5.86% |
|  | LINE1 | 12,130 | 3,357,131 bp | 0.14% |
|  | LINE2 | 100,340 | 30,227,268 bp | 1.23% |
|  | L3/CR1 | 8,437 | 3,557,603 bp | 0.14% |
| LTR elements: |  | 211,377 | 145,957,516 bp | 5.95% |
|  | ERVL | 6 | 360 bp | 0.00% |
|  | ERVL-MaLRs | 0 | 0 bp | 0.00% |
|  | ERV_classI | 19,402 | 1,295,486 bp | 0.05% |
|  | ERV_classII | 5,672 | 1,603,072 bp | 0.07% |
| DNA elements: |  | 1,603,010 | 421,567,495 bp | 17.18% |
|  | hAT-Charlie | 32,085 | 3,719,809 bp | 0.15% |
|  | TcMar-Tigger | 52,635 | 2,0922,832 bp | 0.85% |
| Unclassified: |  | 2,158,454 | 668,949,483 bp | 27.26% |
| Total interspersed repeats: |  |  |  |  |
| Small RNA: |  | 52,314 | 10,622,911 bp | 0.43% |
| Satellites: |  | 15,462 | 4,216,424 bp | 0.17% |
| Simple repeats: |  | 40,358 | 10,377,834 bp | 0.42% |
| Low complexity: |  | 149 | 24,863 bp | 0.00% |

orthologous databases used (Table 3). The number of functionally annotated genes was also higher than those of Genome V1, with 4,065 additional genes annotated (Tables 4, 6 and 7). Overall, the numbers of both predicted and annotated genes are within the expected range for bivalves (reviewed in [4]), as well as within the records of other freshwater mussel assemblies [26, 53].

## CONCLUSION

In this report, a new and highly improved genome assembly for the freshwater pearl mussel is presented. This genome assembly, produced using PacBio long-read sequencing, significantly improves contiguity without scaffolding. Unlike other freshwater mussels' genomes, the one presented here has not been scaffolded (i.e., it has no gaps of undetermined size), thus representing an ideal framework to employ chromosome anchoring approaches, such as Hi-C sequencing. This new genome represents a key resource to start exploring the many biological, ecological, and evolutionary features of this highly threatened group of organisms, for which the availability of genomic resources still falls far behind other molluscs.

## DATA AVAILABILITY

All software with respective versions and parameters used for producing the resources presented here (i.e., transcriptome assembly, pre- and post-assembly processing stages, and transcriptome annotation) are listed in the methods section. Software programs with no parameters associated were used with the default settings.

The raw sequencing reads were deposited at the National Center for Biotechnology Information (NCBI) Sequence Read Archive with the accession numbers SRR23176563 (Illumina PE) and SRR23176561 (PacBio CLR). The new genome assembly is also available

**Table 6.** Structural annotation report of the new *M. margaritifera* genome assembly.

| Structural annotation | Number |
|---|---|
| Number of genes | 40,165 |
| Number of mRNAs | 48,314 |
| Number of CDSs | 48,314 |
| Number of exons | 328,489 |
| Number of introns | 280,173 |
| Number of start_codons | 48,314 |
| Number of stop_codons | 48,314 |
| Number of exons in CDSs | 328,489 |
| Number of introns in CDSs | 280,175 |
| Number of introns in exon | 280,175 |
| Number of introns in intron | 240,261 |
| Number gene overlapping | 440 |
| Number of single exon gene | 8,092 |
| Number of single exon mRNAs | 8,402 |
| Mean mRNAs per gene | 1.2 |
| Mean CDSs per mRNA | 1.0 |
| Mean exons per mRNA | 6.8 |
| Mean introns per mRNA | 5.8 |
| Mean exons per CDS | 6.8 |
| Mean introns in CDSs per mRNA | 5.8 |
| Mean introns in exons per mRNA | 5.8 |
| Mean introns in introns per mRNA | 5.0 |
| Total gene length | 1,134,996,674 |
| Total mRNA length | 1,399,972,668 |
| Total CDS length | 65,168,232 |
| Total exon length | 65,168,232 |
| Total intron length | 1,334,804,372 |
| Total start_codon length | 144,942 |
| Total stop_codon length | 144,942 |
| Total intron length per CDS | 1,334,804,436 |
| Total intron length per exon | 1,334,804,436 |
| Total intron length per intron | 38,816,274 |
| Mean gene length | 28,258 |
| Mean mRNA length | 28,976 |
| Mean CDS length | 1,348 |
| Mean exon length | 198 |
| Mean intron length | 4,764 |
| Mean CDS piece length | 198 |
| Mean intron in CDS length | 4,764 |
| Mean intron in exon length | 4,764 |
| Mean intron in intron length | 161 |
| Longest gene | 492,278 |
| Longest mRNA | 492,278 |
| Longest CDS | 50,892 |
| Longest exon | 14,931 |
| Longest intron | 270,677 |
| Longest CDS piece | 14,931 |
| Longest intron into CDS part | 270,677 |
| Longest intron into exon part | 270,677 |
| Longest intron into intron part | 14,931 |
| Shortest gene | 123 |
| Shortest mRNA | 123 |
| Shortest CDS | 9 |
| Shortest exon | 3 |
| Shortest intron | 33 |

**Table 7.** Functional annotation report of *M. margaritifera* genome assembly.

| Functional annotation | Number |
|---|---|
| Swissprot/RefSeq | 21,050 |
| CDD | 11,151 |
| Coils | 7,148 |
| GO | 18,024 |
| Gene3D | 23,613 |
| Hamap | 383 |
| InterPro | 29,030 |
| KEGG | 1,495 |
| MetaCyc | 1,445 |
| MobiDBLite | 12,555 |
| PIRSF | 973 |
| PRINTS | 5,250 |
| Pfam | 25,322 |
| ProSitePatterns | 6,471 |
| ProSiteProfiles | 15,115 |
| Reactome | 5,837 |
| SFLD | 110 |
| SMART | 12,494 |
| SUPERFAMILY | 23,283 |
| TIGRFAM | 1,148 |
| Total | 34,137 |

on NCBI under the accession number JAQPZY000000000. The BioSample accession number is SAMN32798282, and the BioProject one is PRJNA925505. All the remaining data has been uploaded to figshare [54], including the final unmasked and masked genome assemblies (Mma.fa and Mma_SM.fa), the annotation file (Mma_annotation_v1.gff3), the predicted genes (Mma_genes_v1.fasta), the predicted messenger RNA (Mma_mrna_v1.fasta), the predicted open reading frames (Mma_cds_v1.fasta), the predicted proteins (Mma_proteins_v1.fasta), as well as the full table reports for the Braker gene predictions, the InterProScan functional annotations (Mma_annotation_v1_InterPro_report.txt), and the RepeatMasker predictions (Mma_annotation_v1_RepeatMasker.tbl). Data supporting this work are openly available in the GigaDB repository [55].

## LIST OF ABBREVIATIONS

AGAT: Another Gtf/Gff Analysis Toolkit; BUSCO: Benchmarking Universal Single-Copy Ortholog; BWA: Burrows-Wheeler Aligner; CDS: Coding sequences; KAT: K-mer analyses toolkit; NCBI: National Center for Biotechnology Information; PacBio: Pacific Biosciences; PE: paired-end; QUAST: Quality Assessment Tool for Genome Assemblies; SMRT: Single Molecule, Real-Time; WGS: whole genome sequencing.

## DECLARATIONS

### Ethical approval

This work has been approved by the CIIMAR ethical committee and by CIIMAR Managing Animal Welfare Body (ORBEA) according to the European Union Directive 2010/63/EU.

### Competing Interests

The authors declare that they have no competing interests.

## Authors' contributions

EF, MLL, and LFCC designed and conceived this work. MLL, VP, and AT were responsible for the field sampling. AGS and AMM carried out the bioinformatics analyses. LFCC, EF, GA, and TF revised the bioinformatics analyses. AGS and EF wrote the first version of the manuscript. All authors read, revised, and approved the final manuscript.

## Funding

AGS was funded by the Portuguese Foundation for Science and Technology (FCT) under the grant SFRH/BD/137935/2018 and COVID/DB/152933/2022, which also supported MLL (2020.03608.CEECIND) and EF (CEECINST/00027/2021). This research was developed under the project EdgeOmics - Freshwater Bivalves at the Edge: Adaptation genomics under climate-change scenarios (PTDC/CTA-AMB/3065/2020) funded by FCT through national funds. Additional strategic funding was provided by FCT UIDB/04423/2020 and UIDP/04423/2020.

## Acknowledgements

The authors thank the two reviewers for the helpful remarks and suggestions, which have significantly improved the manuscript.

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
