## [Reviewer Report]

Reviewer name and names of any other individual's who aided in reviewer Jin SunDo you understand and agree to our policy of having open and named reviews, and having your review included with the published papers. (If no, please inform the editor that you cannot review this manuscript.)YesIs the language of sufficient quality?YesPlease add additional comments on language quality to clarify if needed
Are all data available and do they match the descriptions in the paper? YesAdditional CommentsAre the data and metadata consistent with relevant minimum information or reporting standards? See GigaDB checklists for examples <a href="http://gigadb.org/site/guide" target="_blank">http://gigadb.org/site/guide</a>YesAdditional CommentsIs the data acquisition clear, complete and methodologically sound?YesAdditional CommentsIs there sufficient detail in the methods and data-processing steps to allow reproduction?YesAdditional CommentsIs there sufficient data validation and statistical analyses of data quality? YesAdditional CommentsIs the validation suitable for this type of data?YesAdditional CommentsIs there sufficient information for others to reuse this dataset or integrate it with other data?NoAdditional CommentsAny Additional Overall Comments to the AuthorGomes-dos-Santos et al., have upgraded the freshwater mussel Margaritifera margaritifera genome with the usage of long-read sequencing. Overall, this version has been dramatically improved compared to the former one, with the increased N50 value and BUSCO score and decreased No. of contigs. Considering the important economic value of M. margaritifera and the high quality of assembly, I must congratulate the authors on this. However, in contrast to the high-quality assembly, I am a bit aware of the genome annotation part. To me, the number of gene models predicted is a bit higher compared with other molluscan genomes. This can also be reflected by the low proportion of gene models that can be annotated by Swissprot or GO etc. I suspect that the high number of gene models could be the consequence that only the ab initio evidence was applied in the current study. More sophisticated ways, such as EVM or maker, shall be used to see whether the number of gene models can be reduced without sacrificing the BUSCO scores on the gene models.   Line 76, The official name shall be “Oxford Nanopore Technology (ONT)”.  Fig. 1, it is interesting to see the wide distribution of M. margaritifera. I am a bit interested to know whether there are any genetic differentiations between the European population and the North American population. 
RecommendationMinor Revision

---

## [Reviewer Report]

Reviewer name and names of any other individual's who aided in reviewer Rebekah RogersDo you understand and agree to our policy of having open and named reviews, and having your review included with the published papers. (If no, please inform the editor that you cannot review this manuscript.)YesIs the language of sufficient quality?YesPlease add additional comments on language quality to clarify if needed
Are all data available and do they match the descriptions in the paper? YesAdditional CommentsAre the data and metadata consistent with relevant minimum information or reporting standards? See GigaDB checklists for examples <a href="http://gigadb.org/site/guide" target="_blank">http://gigadb.org/site/guide</a>YesAdditional CommentsIs the data acquisition clear, complete and methodologically sound?YesAdditional CommentsIs there sufficient detail in the methods and data-processing steps to allow reproduction?YesAdditional CommentsAll methods seem standard and high quality for a genome release.Is there sufficient data validation and statistical analyses of data quality? YesAdditional CommentsIf the authors could add a table comparing with other Unio genomes, that might be helpful. Gene numbers, busco scores, N50s, and other relevant stats. It will help readers see the value of this more contiguous genome -V. ellipsiforma (Renaut et al.)  -M nervosa  -P. streckersonii 
Is the validation suitable for this type of data?YesAdditional CommentsIs there sufficient information for others to reuse this dataset or integrate it with other data?YesAdditional CommentsAny Additional Overall Comments to the AuthorRecommendationAccept